# Unidirectional versus bidirectional brushing: Simulating wind influence on *Arabidopsis thaliana*

**Original Research Article**

*Arabidopsis thaliana*; biomechanics; mechanical properties; thigmomorphogenesis; tropic response.

**Author for correspondence:**
Oleksandr Zhdanov
Email: Oleksandr.Zhdanov@glasgow.ac.uk

Oleksandr Zhdanov[1,2] , Michael R. Blatt[2] , Hossein Zare-Behtash[1] and Angela Busse[1]

[1]James Watt School of Engineering, University of Glasgow, Glasgow G12 8QQ, United Kingdom; [2]Laboratory of Plant Physiology and Biophysics, Bower Building, University of Glasgow, Glasgow G12 8QQ, United Kingdom

## Abstract

Plants acclimate to various types of mechanical stresses through thigmomorphogenesis and alterations in their mechanical properties. Although resemblance between wind- and touch-induced responses provides the foundation for studies where wind influence was mimicked by mechanical perturbations, factorial experiments revealed that it is not always straightforward to extrapolate results induced by one type of perturbation to the other. To investigate whether wind-induced changes in morphological and biomechanical traits can be reproduced, we subjected *Arabidopsis thaliana* to two vectorial brushing treatments. Both treatments significantly affected the length, mechanical properties and anatomical tissue composition of the primary inflorescence stem. While some of the morphological changes were found to be in line with those induced by wind, changes in the mechanical properties exhibited opposite trends irrespective of the brushing direction. Overall, a careful design of the brushing treatment gives the possibility to obtain a closer match to wind-induced changes, including a positive tropic response.

## 1 Introduction

Plants respond to mechanical perturbations through morphological and physiological changes termed 'thigmomorphogenesis' by Jaffe (1973). The most common responses found in many plant species are inhibition of the stem length, increase in stem radial growth and redistribution of biomass from above- to below-ground (Chehab et al., 2009; Telewski, 2016). Other reported responses include alterations in flowering time, chlorophyll content, number of leaves, senescence and development of stress resistance (see Biddington, 1986). In addition, plants alter their mechanical properties to cope with the induced mechanical stress. However, the direction of this response, for example, whether a plant stem becomes more rigid or compliant, is not universal and depends on the type of perturbation and the plant genotype (Pruyn et al., 2000; Telewski, 2016; Telewski & Jaffe, 1986b).

Initially, thigmomorphogenesis was recorded as a result of the application of artificial mechanical stress in the form of rubbing (Jaffe, 1973). Other types of mechanical perturbations, such as bending or flexing (Coutand et al., 2010; Pruyn et al., 2000), brushing (Paul-Victor & Rowe, 2011) and shaking (Niklas, 1998), which also involve solid-to-solid contact (e.g., brushing material to plant), evoke similar responses in plants and have been widely used in thigmomorphogenetic studies.

In their natural environment, terrestrial plants regularly experience perturbations from various natural sources, such as wind, rain, snow and animals; among which wind is widely considered as the major one. Plants acclimate to windy environments through morphological changes that are similar to thigmomorphogenetic response resulting from artificial mechanical perturbations (Biddington, 1986; Gardiner et al., 2016; Jaffe & Forbes, 1993), although wind exerts perturbations through fluid (air) to solid (plant) interaction. Based on these similarities, various types of mechanical perturbations have been applied to plants to mimic the effects of wind (e.g., Niez et al., 2019; Niklas, 1998; Paul-Victor & Rowe, 2011).

However, the results of factorial experiments suggest that wind and mechanical perturbations can have significantly different effects on the same plants when applied separately. Smith and Ennos (2003) showed that while exposure of sunflowers (*Helianthus annuus* L.) to airflow resulted in taller plants with less rigid stems, mechanical flexure had the opposite effects. Similarly, the effects of wind and brushing on *Plantago major* L. were shown to be opposite in terms of the morphology of petioles and laminas (Anten et al., 2010). Consequently, as concluded by Anten et al. (2010), it is not always correct to extrapolate the effects of mechanical perturbations to the effects of wind.

In our recent study (Zhdanov et al., 2021), we demonstrated that the response of the widely used model plant Arabidopsis to a constant unidirectional wind is significantly different compared with those reported as a result of brushing (Paul-Victor & Rowe, 2011) in terms of changes in plant morphology, mechanical properties and anatomical tissue composition of its primary inflorescence stem. Moreover, Arabidopsis exhibits a positive anemotropic response to this type of treatment. A possible source of these discrepancies could be the difference in the direction of the applied stress, that is, the wind treatment in Zhdanov et al. (2021) was unidirectional, while Paul-Victor and Rowe (2011) applied bidirectional brushing.

Based on these observations, it can be hypothesised that a careful design of the brushing experiment, especially in terms of the direction of applied perturbations, could allow for a closer mimicking of a unidirectional wind treatment and, hence, the observed response of Arabidopsis, including a positive tropic response, will be similar. The aim of the present study is to investigate the possibility to mimic the effects of a constant unidirectional wind through mechanical perturbations. To address this question, Arabidopsis ecotype Col-0 was subjected to two types of vectorial brushing treatment, namely bidirectional and unidirectional, in two separate experiments. In addition, two different brushing materials were applied in each experiment to investigate possible differences in the plant response. The use of two brushing materials is motivated by the variety of materials that have been used in brushing experiments, for example, paper (Wang et al., 2009), cardboard (Latimer, 1990), wooden bar (Keller & Steffen, 1995), duster (Anten et al., 2010) and polythene (Paul-Victor & Rowe, 2011), which have different surface textures and thus could result in different effects on plants. The tropic response and thigmomorphogenetic changes to Arabidopsis morphology, mechanical properties and anatomical tissue composition of the primary inflorescence stem were assessed and compared between and within the experiments.

## 2 Materials and methods

### 2.1 Plants

Seeds of Arabidopsis (ecotype Columbia-0) were sown in a single pot and kept at 4°C for 48 hr. The pot was then placed in the growth chamber with a long-day cycle (16 hr of light and 8 hr of darkness), temperature at 22°C, light intensity at 150 $\mu$mol m$^{-2}$ s$^{-1}$ and humidity at 60%. After approximately 2 weeks, the seedlings were transplanted into individual pots (pot diameter = 76 mm) and moved into the growth room where the experiments were conducted. The conditions in the growth room were as follows: long-day cycle, temperature at 19°C and light intensity at 150 $\mu$mol m$^{-2}$ s$^{-1}$. After approximately 20–25 days, when the flower bearing stem started to develop, plants were randomly separated into three groups. Two groups (20 plants each) were subjected to mechanical perturbations (Experimental Groups 1 and 2) by two different

brushing materials. The third group (20 plants) was grown on the same shelf in the growth room under the same conditions but without any perturbations (control group). This growth procedure was used in the two sets of brushing experiments conducted in this study which are described in the following subsection.

### 2.2 Mechanical perturbations

Mechanical perturbations in the form of brushing were performed by a bespoke brushing machine (Figure 3a–c). The machine has two belt-driven linear actuators which enable movements along horizontal and vertical axes (Figure 3a). The actuators are controlled by individual step motors with controllers allowing for separate and fully automated operation based on a preset programme. A beam mounted on the belt-driven gantry of the vertical axis is used for attaching the brushing material for the experiments. The beam extends on both sides of the machine and thus allows to use two different brushing materials in a single experiment.

In the present study, two sets of brushing experiments were carried out. Each set of experiments contained two experimental groups and one control group. The experimental groups were placed on opposite sides of the machine along its horizontal axis and subjected to brushing by different materials for 2 weeks. Plants from Experimental Group 1 were brushed by a textured jute fabric, whereas plants from Experimental Group 2 were brushed with smooth plastic. The brushing elements were made by combining plastic sheets with jute fabric pieces of the same size and folding them with one or the other side up. This ensured they had the same mass, and the plants were brushed with equal force. Consequently, any morphological changes or alterations observed in the mechanical properties, were expected to be solely a result of differences in the surface texture of the brushing material.

In the first set of experiments, plants were subjected to bidirectional brushing (Figure 3b). Every 4 hr, the plants in both experimental groups were brushed 20 times (10 times in each direction). In the second set of experiments, unidirectional brushing was applied (Figure 3c): in this case, plants were subjected to the same number and frequency of mechanical perturbations (20 brushes every 4 hr), but all brushes were made in one direction. In both modes, only the stems and not the rosettes were mechanically perturbed. The height of the beam with the brushing material was adjusted as plants grew, so that the primary inflorescence stems were deflected by 45–65° from their vertical orientation during the brushing.

### 2.3 Phenotyping

Morphological changes in Arabidopsis phenotype as a result of mechanical perturbations were assessed at the end of each experiment. For each plant in the control and experimental groups, the length of the primary inflorescence stem, the number of stems (basal branches) and the number of branches were determined. The aboveground fresh biomass was assessed for 10 randomly chosen plants from each group. In addition, after oven drying at 70°C, their dry biomass was also determined. For the other 10 plants from each group, the average diameters of their bottom and top parts were evaluated as part of their mechanical characterisation.

### 2.4 Mechanical characterisation

Mechanical characterisation of the primary inflorescence stems of perturbed and control plants was carried out using the dynamic forced vibration method (Zhdanov et al., 2020) for 10 plants from

each group. Two segments of the primary inflorescence stem were characterised. The first segment, referred to as 'bottom part of the stem', was taken from the basal part of the stem. The second segment was taken from the apex part of the stem excluding the growth zone and is referred to as 'top part of the stem'. The tests were conducted directly after the segment under consideration was cut from the stem using a razor blade and, if required, cleared from branches, fruits, flowers and young floral buds. This eliminated the influence of turgor pressure reduction and dehydration on the determined mechanical properties. In addition, at the time of the tests, none of the stems showed any sign of senescence that could also affect their mechanical properties.

The dynamic forced vibration method allows to determine multiple resonant frequencies ($f_i$) of the tested stem segment that are related to its mechanical properties through Euler–Bernoulli beam theory (Blevins, 1979):

$$f_i = \frac{\lambda_i^2}{2\pi L^2}\sqrt{\frac{EI}{m}}, \quad i = 1,2,3,\ldots,n, \tag{1}$$

where $E$ is the Young's modulus of elasticity which characterises the ability of the material to resist elastic deformations, $L$ is the length of the stem, $I$ is the second moment of area, $m$ is the mass per unit length and $\lambda_i$ is a dimensionless parameter that is obtained from the characteristic equation corresponding to the vibration mode and applied boundary conditions. The stem segments were tested as clamped–clamped beams, and up to four vibration modes were considered with the corresponding values of $\lambda_i$ from Blevins (1979; $\lambda_1 = 4.73004074$, $\lambda_2 = 7.85320462$, $\lambda_3 = 10.9956079$ and $\lambda_4 = 14.1371655$). The detailed testing procedure can be found in Zhdanov et al. (2020). The product of $EI$ is known as bending rigidity and characterises the ability of the structure to resist bending. In the present study, both $E$ and $EI$ were determined for each tested stem segment.

The cross section of Arabidopsis was approximated as a circle (Bichet et al., 2001; Turner & Somerville, 1997), thus $I$ was calculated as:

$$I = \frac{\pi}{64}D^4, \tag{2}$$

where $D$ is the diameter of the tested stem segment that was determined as an average value over several measurements of the stem diameter at different locations along its length. The measurements were done post hoc using ImageJ (Schneider et al., 2012) from the photographs of the tested stem segments taken after each test. The mass of the tested segments was also measured after each test using a precision balance to evaluate $m$, which is required to estimate the mechanical properties using equation (1).

### 2.5 Anatomical measurements

Anatomical tissue organisation was observed on the next day after mechanical characterisation for the same stem segments. Prior to this, the stem segments were stored in individual falcon tubes filled with distilled water at 4°C. Manually sectioned transverse segments from the central parts of each stem segment were stained with 0.02% toluidine blue. Histochemical staining enabled visual differentiation between the structural tissues of the studied segments. The samples were observed on a Zeiss Stemi SV11 microscope, and photographs were captured. The relative areas of the outer part (epidermis and cortex), the middle part (lignified tissues coloured in blue) and the innermost part (purple coloured pith) were measured from the images using the ImageJ software.

### 2.6 Statistical analysis

All statistics of measured quantities are reported as mean ± standard deviation of $n$ observations. A parametric one-way ANOVA test with post hoc Tukey test was employed for statistical analysis. The tests were performed in MATLAB (R2020a, MathWorks, Natick, MA, USA) using in-built functions. A statistically significant difference was established at $p \leq .05$.

## 3 Results

Due to the availability of only one brushing machine, two sets of experiments were conducted to explore the influence of vectorial brushing treatment on Arabidopsis in series. Additional experiments with own control groups were conducted following the same procedure and using the same configuration for brushing directionality and brushing material. In the interests of brevity, the results of these experiments are not presented in the paper but included as Supplementary Material 3. Although there are some quantitative differences between values of studied parameters, the effects of brushing treatments relative to the respective control groups are reproducible and lead to the same conclusions.

### 3.1 Arabidopsis morphology

Both bi- and unidirectional types of brushing treatment resulted in a statistically significant ($p < .0001$) inhibition of the primary inflorescence stem length (Figure 1a). The reduction of the stem length appears to be dependent on the applied type of brushing. Compared with the control group, plants subjected to the bidirectional brushing, on average, were shorter by 25%, whereas in the unidirectional brushing experiment, the average reduction was only 18%. In contrast, no significant changes ($p > .05$) to the number of stems and branches were imposed by any type of the applied brushing treatment, and values of these parameters were similar in the experimental and control plants (Figure 1b,c).

In addition, mechanical perturbations in the form of brushing had only minor effects on the other measured morphological traits (Table 1). While the brushing treatment resulted in a reduction of the diameter of the bottom part of the stem, this reduction was marginal and, in most cases, not statistically significant ($p > .05$) compared to the control group, except for Experimental Group 1 in the bidirectional brushing experiment ($p = .044$). Furthermore, no consistent effect of both types of brushing was observed for the top part of the stem, and the diameters of these segments were similar in all groups. Mechanically perturbed plants also had slightly lower fresh and dry biomass compared to the plants from the control group in both conducted experiments. However, this reduction was not statistically significant ($p > .05$).

No significant difference in the effects induced by different types of brushing material, namely textured jute fabric (Experimental Group 1) and smooth plastic (Experimental Group 2), was observed in both experiments. All recorded morphological parameters had similar values (see Figure 1 and Table 1) irrespective of the used brushing material.

### 3.2 Mechanical properties of primary inflorescence stems

At the time of the mechanical characterisation, the primary inflorescence stems were upright and self-supporting. No signs of damage, inflicted by mechanical perturbations in the form of brushing, were observed in plants from the experimental groups in both

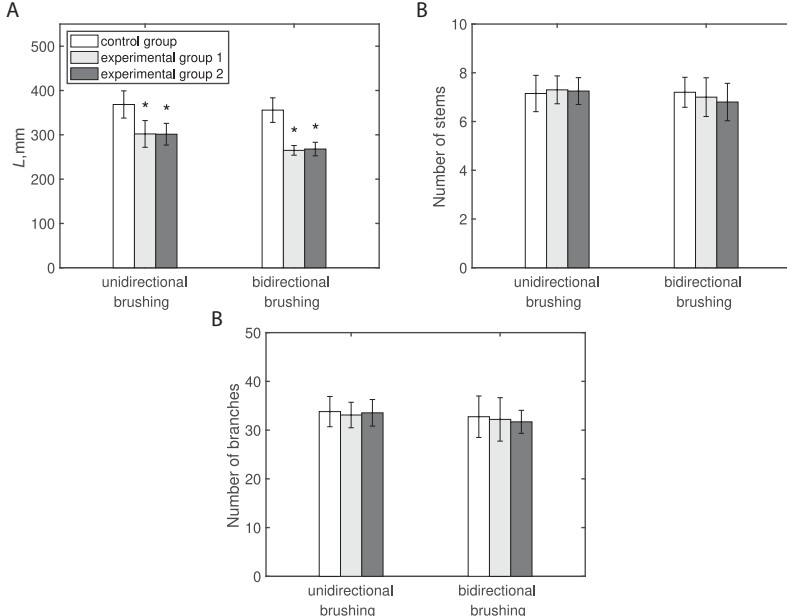

**Fig. 1** The effect of mechanical perturbations by different types of brushing treatment and brushing material on Arabidopsis morphology. (a) Primary inflorescence stem length. (b) Number of stems. (c) Number of branches. The legend in part (a) applies to all parts of this figure. Experimental Group 1 was brushed with textured jute fabric, whereas Experimental Group 2 with a smooth plastic in both experiments. Statistically significant differences ($p \leq .05$) between groups were identified using ANOVA with a post hoc Tukey test.

* marks statistically significant difference compared with the control group in the respective experiment. The raw data for both experiments can be found in Supplementary Material 1.

**Table 1.** The effect of mechanical perturbations by different types of brushing and brushing material on morphological and biomechanical parameters of Arabidopsis

| | Unidirectional brushing | | | Bidirectional brushing | | |
|---|---|---|---|---|---|---|
| | Control | Experimental Group 1 | Experimental Group 2 | Control | Experimental Group 1 | Experimental Group 2 |
| Whole plant | | | | | | |
| Fresh biomass (g) | $10.94 \pm 1.16$ | $10.61 \pm 0.92$ | $10.49 \pm 0.95$ | $10.35 \pm 0.96$ | $9.82 \pm 0.55$ | $9.86 \pm 0.57$ |
| Dry biomass (g) | $1.10 \pm 0.14$ | $1.05 \pm 0.08$ | $1.07 \pm 0.11$ | $1.10 \pm 0.14$ | $1.01 \pm 0.09$ | $1.03 \pm 0.10$ |
| Bottom part of the stem | | | | | | |
| $D$ (mm) | $1.46 \pm 0.11$ | $1.42 \pm 0.11$ | $1.35 \pm 0.15$ | $1.73 \pm 0.13$ | $1.56 \pm 0.12^{*}$ | $1.67 \pm 0.19$ |
| $EI$ (N mm$^2$) | $248 \pm 69$ | $182 \pm 55^{*}$ | $153 \pm 61^{*}$ | $497 \pm 163$ | $208 \pm 84^{*}$ | $273 \pm 105^{*}$ |
| Top part of the stem | | | | | | |
| $D$ (mm) | $1.04 \pm 0.08$ | $1.04 \pm 0.10$ | $1.00 \pm 0.07$ | $1.06 \pm 0.05$ | $1.05 \pm 0.04$ | $1.04 \pm 0.03$ |
| $EI$ (N mm$^2$) | $33 \pm 10$ | $29 \pm 4$ | $27 \pm 7$ | $34 \pm 6$ | $32 \pm 4$ | $32 \pm 7$ |

*Note:* Experimental Group 1 was brushed with textured jute fabric, whereas Experimental Group 2 with smooth plastic in both experiments. Values are presented as mean $\pm$ SD with $n = 10$ for each case. Statistically significant differences ($p \leq .05$) between groups were identified using ANOVA with a post hoc Tukey test.
* marks statistically significant difference compared with the control group in the respective experiment.

conducted experiments. Unidirectional and bidirectional brushing treatments resulted in significant changes ($p < .01$) to the intensive (independent of object size), and extensive (object size dependent), mechanical properties of the bottom part of the primary inflorescence stem. Compared to the control group, the unidirectional brushing decreased $E$ of this part of the stem by approximately 20%, while a reduction of more than 35% was observed due to bidirectional brushing (Figure 2a). The $EI$ of the bottom parts of the perturbed plants was approximately half of the values determined for the control group in both conducted experiments (Table 1).

In contrast, the Young's modulus of elasticity (Figure 2b) and bending rigidity (Table 1) of the top parts of the stems were not significantly ($p > .05$) affected by any type of the applied brush-

ing treatments. In addition, no significant differences ($p > .05$) in the mechanical properties were found within a single experiment between plants from experimental groups brushed with either smooth plastic or textured jute fabric. For both tested stem segments, the values of $E$ and $EI$ had similar values irrespective of the brushing material.

### 3.3 Anatomical tissue composition

Both types of vectorial brushing treatment resulted in significant changes to the anatomical tissue organisation of the bottom part of the primary inflorescence stem of Arabidopsis (Table 2). Brushed plants developed proportionally more cortex and epidermis

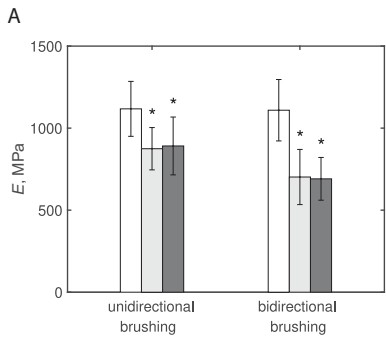 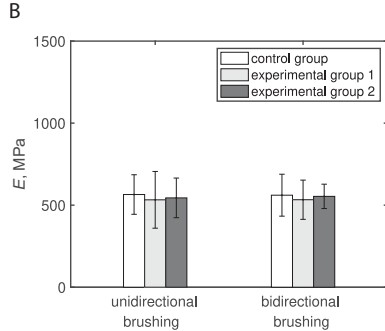

**Fig. 2** The effect of mechanical perturbations by different types of brushing treatment and brushing material on the intensive mechanical properties of Arabidopsis primary inflorescence stem. (a) Modulus of elasticity of the bottom part of the stem. (b) Modulus of elasticity of the top part of the stem. The legend in part (b) applies to both parts of this figure. Experimental Group 1 was brushed with textured jute fabric, whereas Experimental Group 2 with smooth plastic in both experiments. Statistically significant differences ($p \leq .05$) between groups were identified using ANOVA with a post hoc Tukey test.
* marks statistically significant difference compared with the control group in the respective experiment. The raw data for both experiments can be found in Supplementary Material 2.

**Table 2.** The effect of mechanical perturbations by different types of brushing and brushing material on the contribution of anatomical tissues to the total cross-sectional area of the primary inflorescence stem of Arabidopsis

| | Unidirectional brushing | | | Bidirectional brushing | | |
|---|---|---|---|---|---|---|
| | Control | Experimental Group 1 | Experimental Group 2 | Control | Experimental Group 1 | Experimental Group 2 |
| **Bottom part of the stem** | | | | | | |
| Pith | $43.69 \pm 1.73$ | $45.45 \pm 1.92$ | $44.93 \pm 2.93$ | $44.19 \pm 1.76$ | $45.66 \pm 1.72$ | $44.73 \pm 4.49$ |
| Lignified tissues | $30.14 \pm 1.17$ | $24.07 \pm 1.29^*$ | $23.42 \pm 2.15^*$ | $29.55 \pm 1.68$ | $23.15 \pm 2.57^*$ | $24.17 \pm 3.06^*$ |
| Cortex and epidermis | $26.16 \pm 1.78$ | $30.48 \pm 1.95^*$ | $31.65 \pm 2.10^*$ | $26.26 \pm 1.46$ | $31.19 \pm 2.46^*$ | $31.10 \pm 3.19^*$ |
| **Top part of the stem** | | | | | | |
| Pith | $42.28 \pm 1.29$ | $42.58 \pm 1.14$ | $42.32 \pm 1.84$ | $44.06 \pm 1.21$ | $43.12 \pm 1.38$ | $42.89 \pm 1.94$ |
| Lignified tissues | $23.36 \pm 1.51$ | $22.38 \pm 1.50$ | $22.54 \pm 1.42$ | $23.36 \pm 1.28$ | $22.82 \pm 1.29$ | $23.57 \pm 1.99$ |
| Cortex and epidermis | $34.36 \pm 1.73$ | $35.03 \pm 1.51$ | $35.14 \pm 1.42$ | $32.58 \pm 1.36$ | $34.05 \pm 1.39$ | $33.62 \pm 1.33$ |

*Note:* Experimental Group 1 was brushed with textured jute fabric, whereas Experimental Group 2 with a smooth plastic in both experiments. Values are presented as mean $\pm$SD with $n = 10$ for each case. Statistically significant differences ($p \leq .05$) between groups were identified using ANOVA with a post hoc Tukey test.
* marks statistically significant difference compared with the control group in the respective experiment.

compared to the control group. In addition, a reduction of the relative area of lignified tissues was observed in the plants from both experimental groups. In both cases, the observed alterations are statistically significant ($p < .001$). However, the brushing treatment did not show significant effects on pith in this part of the stem.

Similar to the other aforementioned parameters that were recorded for the top part of the stem, its anatomical tissue organisation was not affected either. All relative areas of the considered tissues had similar values in all groups (Table 2). Moreover, no significant differences were found between tissue contributions due to different texture of the brushing material in both experiments.

# 4 Discussion

## 4.1 Morphological changes

The observed inhibition of the stem length due to different types of mechanical perturbations is consistent with the previously reported results for various herbaceous and woody plants, such as *Brachypodium distachyon* (Gladala-Kostarz et al., 2020), *Helianthus annuus* (Smith & Ennos, 2003), *Zea mays, Cucumis sativus* (Jaffe, 1973), *Pinus taeda* (Telewski & Jaffe, 1986b) and *Abies fraseri*

(Telewski & Jaffe, 1986a). Moreover, a reduction of the primary inflorescence stem length by approximately 50% was observed in an earlier study where Arabidopsis was subjected to a bidirectional brushing treatment (Paul-Victor & Rowe, 2011). On the one hand, the decrease of the stem length in the present bidirectional brushing experiment is in line with these results; however, the inhibition relative to the control group is lower, namely 25%. This difference can be related to the higher total number of perturbations received by Arabidopsis plants in the study by Paul-Victor and Rowe (2011), where 80 additional brushes were made at the end of each day. In addition, Paul-Victor and Rowe (2011) started the mechanical perturbation of Arabidopsis at an earlier developmental stage, that is, at the leaf developmental stage, whereas in the present study, brushing treatment commenced from the inflorescence emergence stage, to be consistent with the previously conducted wind treatment experiments (Zhdanov et al., 2021). On the other hand, the inhibition of the stem length in the present unidirectional brushing experiment is found to be similar to the effects of exposure to constant unidirectional wind (Zhdanov et al., 2021) and periodic multidirectional wind (Bossdorf & Pigliucci, 2009) of 5 m/s, where stem length reductions of 14 and 13.2% were reported, respectively.

Unlike many other plants (see, e.g., Biddington, 1986; Coutand et al., 2010; Telewski & Jaffe, 1986a), for which increase in the stem radial growth is a common thigmomorphogenetic response to

various types of mechanical stress including wind, both types of the applied brushing treatment decreased the diameter of the bottom part of the stem. However, these changes were not significant in most cases. It should be noted that in the majority of the reported cases of stimulation of radial growth, bending stress was applied to plants with established secondary cambial growth that accounted for the observed thickening. An Arabidopsis stem, in contrast, is composed mainly of primary tissue with only limited presence of interfascicular cambium at its base (Sehr et al., 2010), thus providing an explanation for the observed results. A marginal reduction of the stem radial growth was also reported for Arabidopsis as a result of bidirectional brushing (Paul-Victor & Rowe, 2011). Studies of wind influence on Arabidopsis also reported reduction in the radial growth of the bottom part of the stem (Bossdorf & Pigliucci, 2009; Zhdanov et al., 2021), but these changes were statistically significant. No changes to the diameter of the top part of the stem due to the applied mechanical perturbations were recorded in the present study. This can be attributed to the fact that during the brushing treatment, the stem was bent in its bottom part, whereas the top part was subjected solely to brushing, that is, brushing material was dragged along this part of the stem. This observation is consistent with wind-induced effects reported for the same part of the stem (Zhdanov et al., 2021).

No alterations to the number of stems and branches were observed by any type of applied brushing treatment. These parameters were not reported in Paul-Victor and Rowe (2011), but from the presented photographs, the number of branches in the experimental and control groups appears to be similar. However, at this developmental stage of the plants, the number of branches is quite low, and for possible effects, longer experiments would be required. In case of the constant unidirectional wind treatment (Zhdanov et al., 2021), a significant reduction in both of these parameters was observed, which was linked to the acclimation strategy of Arabidopsis to reduce experienced wind loadings through the reduction of its frontal area. In case of a brushing treatment, this does not apply, and thus Arabidopsis would not benefit from the reduction of the number of stems and branches.

Mechanical perturbations are also known to affect the aboveground biomass due to its reallocation to the belowground part of the plant (Coutand et al., 2008; Kern et al., 2005). The significant decrease in Arabidopsis aboveground biomass has also been reported in the literature as a result of wind treatment (Bossdorf & Pigliucci, 2009; Zhdanov et al., 2021). However, none of the applied types of brushing treatment resulted in significant effect on the biomass in the present study, with only slight reduction observed in all experimental groups. There are several factors that can explain these observations. First, as mentioned above, the brushing treatment was applied when the primary inflorescence stem started to emerge. At this developmental stage, the significant part of the rosette has already been formed (Boyes et al., 2001). Second, the rosettes were not affected by the brushing treatment, since with the present configuration of the brushing material, this could result in the irreversible damage to the primary inflorescence stems. Finally, brushing had no significant effect on the number of stems and branches and the diameter of the primary inflorescence stem; hence, the rest of the aboveground biomass, apart from rosette, was affected only slightly. However, based on the photographs of Arabidopsis after brushing treatment in Paul-Victor and Rowe (2011), it can be presumed that an application of mechanical perturbations at an earlier developmental stage might result in the significant decrease of the aboveground biomass. Furthermore, a different configuration of brushing materials, which would allow to brush

rosette together with the stem, could result in the changes to this parameter.

Overall, brushing of Arabidopsis can reproduce effects of wind on the length of the primary inflorescence stem and partially on the alterations to the stem radial growth and plant biomass. Furthermore, match in the wind and brushing directions gives closer match in the inhibition of the primary inflorescence stem. However, the brushing treatment does not mimic wind-induced changes to the number of stems and branches.

### 4.2 Changes to mechanical properties and anatomical tissue organisation

The observed reduction in the bending rigidity and Young's modulus of elasticity of the stem is consistent with the previous findings in plants subjected to mechanical stress, for example, Anten et al. (2010) and Telewski and Jaffe (1986b). However, it should be noted that changes to the mechanical properties are plant specific and also vary between different genotypes of the same plants (see, e.g., Pruyn et al., 2000). Consequently, it is more informative to compare changes in the mechanical properties to those reported for Arabidopsis Col-0.

The decrease in the bending rigidity and Young's modulus of elasticity was reported for this Arabidopsis ecotype in the result of the bidirectional brushing experiment by Paul-Victor and Rowe (2011); however, the decrease in the mechanical properties observed in the present study is lower. Paul-Victor and Rowe (2011) attributed the reduction in the mechanical properties to the inhibition of the net development due to perturbations in the form of brushing. The evidence for this can be found in the present results by comparing values of the elastic modulus between the present experiments and those reported in Paul-Victor and Rowe (2011): while the values are similar for the bottom part of the stem in the control groups in both studies, the value of $E$ in the experimental group of Paul-Victor and Rowe (2011) is similar to those for the top part in the present study. Thus, it can be suggested that the reduction in the $E$ and $EI$ in the bottom part of the stem is a result of bending due to the brushing treatment and not due to differences in the developmental stages. This also provides an explanation for the negligible effect of the brushing treatment on the top part of the stem, which was subjected to brushing rather than bending. The lower reduction in $E$ in the bottom part of unidirectionally brushed compared to bidirectionally brushed plants found in the present study may be related to the vectorial influence of perturbations, where in the second case, bending of the stem occurred in two directions compared with only a single direction in the first case.

The current results are in line and confirm findings of Paul-Victor and Rowe (2011) that Arabidopsis acclimates to the mechanical perturbations in the form of brushing by producing shorter and more flexible stems. This response is evident from the increase in the stem flexibility through the reduction of its diameter and modulus of elasticity combined with the decrease in the relative area of lignified tissues. These changes allow the stem to bend under the applied loading rather than to resist it by investing into structural tissues to maintain stem orientation during each brushing instance. Similar findings were also reported for other herbaceous plants and plant parts (Anten et al., 2010; Liu et al., 2007) as well as for arborescent (Gibson, 2012) and climbing palms (Isnard et al., 2005; Rowe et al., 2004), that lack secondary vascular cambium, as a mechanism to deal with mechanical stress. In contrast, a constant unidirectional wind treatment resulted in the increase of the

elastic modulus of Arabidopsis (Zhdanov et al., 2021), suggesting a difference between acclimation strategies to wind and brushing treatments which may arise from the difference in the nature of the applied stress (air vs. solid), exposure to it (constant vs. periodic), and exerted force.

The changes to mechanical properties are consistent with the modifications to the anatomical tissue organisation of the primary inflorescence stems of the perturbed plants. A decrease in the relative area of the lignified tissues, which are known to strengthen the cell walls and increase stiffness of Arabidopsis stems (see, e.g., Huang et al., 2001), provides an explanation for the observed decrease in $E$ and consequently $EI$ in the bottom part of the stem. The significant reduction in the area of lignified tissues in brushed Arabidopsis together with an increase of cortex and epidermis was also reported by Paul-Victor and Rowe (2011). In contrast, no changes to the area of pith were observed in the present study, whereas a decrease was found by Paul-Victor and Rowe (2011) due to mechanical perturbations. These observations can also be related to the difference in the developmental rate, since at the earlier growth stages, Arabidopsis has less pith, whose relative area increases as the plant grows. Furthermore, as in the case of the mechanical properties, no significant changes to the anatomical tissue organisation of the top part of the stem due to brushing were found.

Overall, both applied types of brushing treatment induce opposite effects on the modulus of elasticity and relative area of lignified tissues of the Arabidopsis stems compared to wind. Consequently, changes to these parameters cannot be directly extrapolated from experiments where the latter stress is mimicked by the former, at least in the case of constant unidirectional wind.

### 4.3 Tropic response

Depending on the direction of the brushing treatment, Arabidopsis exhibits a tropic response. While no tropic response was observed in the bidirectional brushing experiment (Figure 3d,f), unidirectional brushing evoked a positive tropic response in Arabidopsis (Figure 3e,g), and young seedlings curved in the direction opposite to the direction of brushing. The observed positive thigmotropic response resembles the positive anemotropic response of Arabidopsis to a constant unidirectional wind treatment (Zhdanov et al., 2021), but without a windswept shape of the plant. Therefore, a careful selection of the type of brushing treatment allows to mimic the tropic response in Arabidopsis as in the case of exposure to unidirectional wind, although this response is thigmotropic rather than anemotropic.

In each brushing instance, the stems were deflected from their vertical orientation and returned back to the initial vertical position immediately after. As a result of this inclination, the position of the inflorescence stem with respect to the direction of gravity vector changes and could evoke a gravitropic response. It is widely known that Arabidopsis shoots exhibit negative gravitropism and reorient themselves back to a vertical position within a few hours after permanent inclination (see, e.g., Morita, 2010). Moreover, gravitropic responses in plants can be evoked by temporary changes to their orientation. Caspar and Pickard (1989) and Kiss et al. (1989) estimated the minimum induction time, that is, the time to evoke gravitropic curvature, for Arabidopsis roots to be 30s by extrapolating data obtained for different durations of gravitropic stimulation. Unfortunately, no measurements of this type have been performed for Arabidopsis inflorescence stems. In the present experiments, the gravitropic stimulation due to brushing

was intermittent with the duration in the order of 1s, and it is highly unlikely to result in such curvature of the stem as observed in Figure 3 e,g. In addition, experiments exploring the dose response of plants to gravitropic stimulation usually involve clinorotation to minimise the influence of gravity after stimulation and to isolate the evoked gravitropic response (Perbal et al., 1997), whereas in the present study, after the end of brushing cycle, the plants continued to grow in the vertical positions under constant gravity. Thus, the observed directional growth response of Arabidopsis inflorescence stems as a result of unidirectional brushing treatment is considered to be thigmotropic, that is, due to contact between the brushing material and the plant stem.

A thigmotropic response is known to occur in Arabidopsis roots (see Massa & Gilroy, 2003), but to the best of our knowledge, it has not been reported for its shoots. However, thigmotropic responses were recorded in the aboveground parts of various other plants, for example, the common bean (*Phaseolus vulgaris* L.; Huberman & Jaffe, 1986) and cucumbers (*Cucumis sativus* L.; Takahashi & Jaffe, 1990). For the reviews of thigmotropic responses in plants, see Braam (2005), Jaffe et al. (2002), and Telewski (2012).

The uni- and bidirectional brushing treatments applied to Arabidopsis in the present study resemble those described in Huberman and Jaffe (1986) and Takahashi and Jaffe (1990) where plant parts were rubbed on one or both sides to evoke thigmotropic responses. The resemblance is particularly close for the top part of the stem, which was brushed, that is, the brushing material was dragged along this part of the stem on one or both sides depending on the applied type of treatment, without bending, which occurred in the bottom part of the stem. In case of symmetric stem rubbing on both sides, no thigmotropic response was observed in Takahashi and Jaffe (1990). This observation is similar to the absence of a thigmotropic response in Arabidopsis subjected to bidirectional brushing treatment in the present study. These similarities provide additional evidence that the observed response is thigmotropic. The presence of a thigmotropic response in Arabidopsis gives the possibility to conduct studies focussed on the mechanosensing and thigmotropic response mechanism, since a wide selection of Arabidopsis mutant lines are readily available. In addition, experiments with nonphototropic (Liscum & Briggs, 1995) and shoot gravitropic (Fukaki et al., 1996) mutants could be conducted to investigate the interaction between different tropic responses in Arabidopsis shoots and how these responses affect thigmotropism.

## 5 Conclusions

Artificial vectorial mechanical perturbations in the form of uni- and bidirectional brushing were applied to Arabidopsis to investigate the possibility to mimic the influence of unidirectional wind. The results suggest that some of the changes to morphological parameters and mechanical properties can be reproduced (inhibition of the primary inflorescence stem length and decrease of bending rigidity) or partially reproduced (decrease of stem diameter and biomass) through both types of brushing applied in this study. In contrast, the changes in the modulus of elasticity and relative area of lignified tissues were found to exhibit opposite trends compared with the unidirectional wind treatment. Furthermore, the magnitude of the decrease of $E$ is dependent on the brushing direction, but not on the surface texture of the brushing material. However, all these changes affect only the bottom part of the stem, which is bent during the brushing, while no significant difference between the recorded morphological and biomechanical parameters were

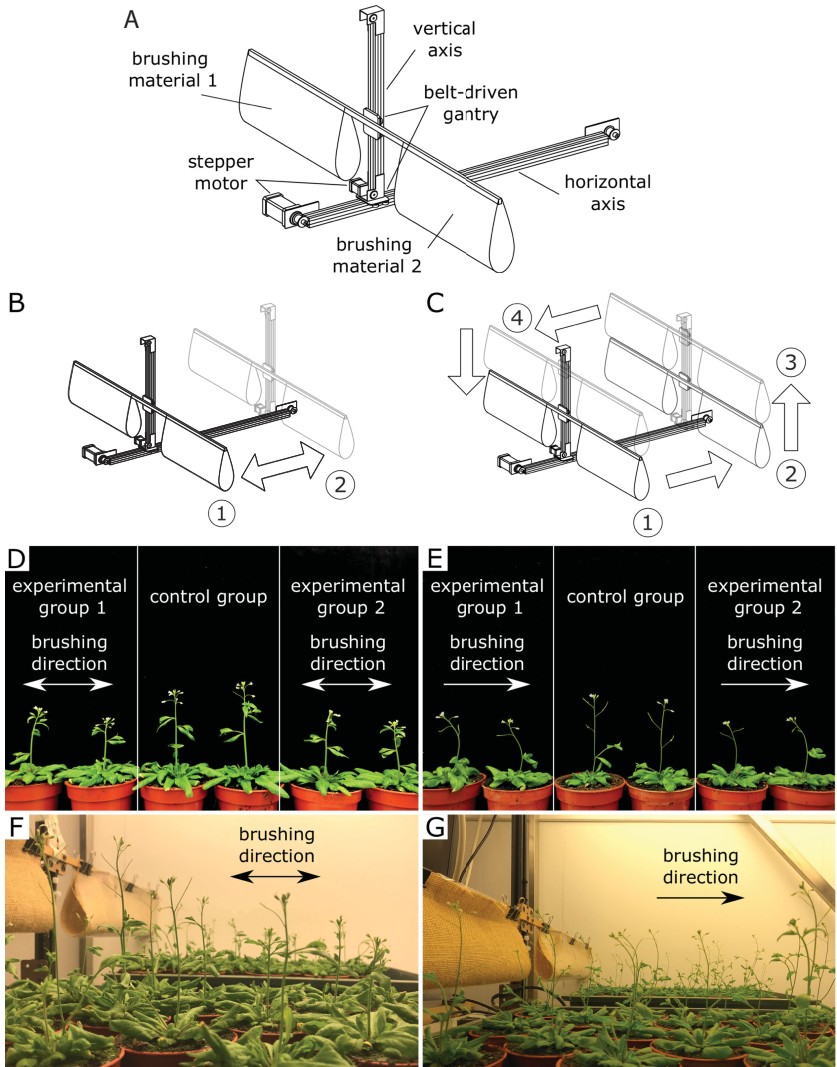

**Fig. 3** Depending on the direction of brushing treatment, it is possible to evoke a tropic response in Arabidopsis similar to that observed as a result of constant unidirectional wind. (a) Schematic diagram of the brushing machine and its components. (b) Bidirectional brushing mode: the plants are brushed in both directions by moving brushing material back and forth (from Position 1 to Position 2). (c) Unidirectional brushing mode: the plants are brushed in a single direction by moving the brushing material from Position 1 to Position 2. Upon reaching the end of the horizontal axis (Position 2), the brushing materials is lifted to Position 3, moved back to the beginning of the axis (Position 4) and returned to the initial Position 1. The plants are placed under the brushing material on both sides of the brushing machine. The position of the beam with brushing material is adjusted in the vertical direction as the plants grow. (d,f) Arabidopsis does not exhibit any tropic response in the bidirectional brushing experiment. (e,g) Arabidopsis exhibits a positive tropic response in the unidirectional brushing experiment. Experimental Group 1 was brushed with a textured jute fabric, whereas a smooth plastic was used as brushing material for Experimental Group 2 in both experiments. The photos were taken on the fourth day after the start of the brushing treatment.

found in the top part of the stem, which was brushed but not bent. Branching of Arabidopsis was not affected by the vectorial type of brushing treatment confirming that reduction in number of branches is an acclimation response to wind which is not evoked by brushing. Arabidopsis exhibited a positive tropic response to unidirectional brushing, which to the best of our knowledge is reported for the first time for the Arabidopsis shoot. This response resembles the anemotropic response to constant unidirectional wind but is considered thigmotropic, since it is a result of brushing that involves solid-to-solid contact.

Overall, brushing treatments can be employed to mimic some aspects of wind influence on Arabidopsis. Moreover, a careful experimental design allows for a closer match between responses to these two types of stresses, including a positive tropic response. However, attention must be paid to the interpretation of the results

of brushing experiments and their extrapolation to wind-induced effects.

## Acknowledgments

This work was supported by the University of Glasgow's Lord Kelvin/Adam Smith (LKAS) PhD Scholarship. We would like to thank Amparo Ruiz-Prado for the help with growing plants and Liam Anderson for the help in the design and assembly of the brushing machine.

**Financial support.** This research received no specific grant from any funding agency, commercial or not-for-profit sectors.

**Conflict of Interest.** The authors declare no conflicts of interest.

**Authorship contributions.** O.Z., M.R.B., H.Z.-B. and A.B. conceptualised and designed the study; O.Z. carried out the experiments, and analysed and visualised the data; O.Z. wrote the draft of the manuscript; M.R.B., H.Z.-B. and A.B. reviewed and edited the manuscript.

**Data availability statement.** The data that support the findings of this study are available from the corresponding author, O.Z., upon reasonable request.

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
