## [Reviewer Report]

Dear Editor,

we would like to submit a manuscript titled “Unidirectional vs bidirectional brushing: simulating wind influence on *Arabidopsis thaliana*” to be considered for publication in the special collection on Plant Biomechanics in Quantitative Plant Biology. 

Various types of mechanical perturbations are widely used for mimicking wind-induced effects on plants. This approach is mainly justified by the similarities in the morphogenetic responses, for example the inhibition of the primary growth, of plants to environmental stresses including wind. However, factorial experiments (Smith and Ennos, 2003; Anten et al., 2010) revealed that it is not always straightforward to extrapolate results induced by one type of perturbation to the other. 

In the present study we applied two types of vectorial brushing treatment, namely uni- and bidirectional, to the model plant *Arabidopsis thaliana* to investigate the possibility to reproduce previously reported effects of constant unidirectional wind (Zhdanov et al., 2020). The changes in plant morphology, mechanical properties, anatomical tissues composition, and tropic response of the primary inflorescence stems were recorded, discussed, and compared to the wind-induced changes. Moreover, a positive thigmotropic response evoked by the unidirectional brushing treatment was recorded for the first time in the aboveground part of Arabidopsis. Overall, while experiments with mechanical perturbations in the form of brushing can reproduce only certain effects of wind, the match in the vectorial component of the induced stress allows for a better agreement with wind-induced changes. 

Sincerely,

---

## [Reviewer Report]

*Comments to Author*: The manuscript “Unidirectional vs bidirectional brushing: simulating wind influence on Arabidopsis thaliana” presents interesting data that complements previous studies by the authors. It is well written but will require some attention to syntax and grammar. The findings help to advance our knowledge about mechanical loading and perturbations on plant growth and development. I have included some minor comments and suggestions to the authors below.

Page 6, Figure A and in results and discussion: I’m not sure what to make of the two controls for the unidirectional and bidirectional treatments. The authors do acknowledge some aspects of the differences on lines 146-148; “Although both experiments were conducted following the same procedure in the same growth room, the differences observed between some parameters are believed to be due to the natural variability.” What natural variability? Within Columbia? They also describe that the two tests were conducted at different times in the growth chamber. It appears that the biggest difference in height growth is the reduced growth of the control plants in the unidirectional brushing test compared to those in the bidirectional test. The heights of the plants exposed to both brushing treatments appear to be about the same size.

Lines 222 – 234. I think it would be helpful here in the discussion about radial growth or in this case, lack there of, for the authors to acknowledge the functional differences between the floral spike and the majority of other plants/organs studied in the cited works in this paragraph. Other than Arabidopsis, I am not aware of any other thigmomorphogenetic studies being conducted on floral spikes. The majority of other studies have been on the flexing of stems of dicot angiosperms, stems of conifers, and stems of monocot angiosperms. With the exception of the monocots, the stems of all other dicots and conifers have some aspect of secondary growth arising from the vascular and cork cambia which results in an increase in radial growth which has been documented to be stimulated by bending/flexing mechanical stress. The inflorescence of Arabidopsis is composed of primary tissue. Cambial tissue is not present in Arabidopsis except for the very base of the floral spike within the foliar rosette. Second, the authors might ponder the function of the floral spike verses the function of a significantly larger stem. The reduction in E appears to be a fairly universal plant tissue/organ response, but an increase in I is not always observed in dicot angiosperms, conifers, and certainly not observed in the primary tissue of the floral spike of Arabidopsis as reported here. Since the floral spike does not obtain a significant height or mass, and is relatively short and somewhat protected in the boundary layer of the ground, the evolved strategy may be to increase flexibility by reducing both E and I along with reducing the cross sectional area of lignified tissues in order to avoid wind loading by bending rather than resist wind loading by increasing cross sectional diameter and the volume of lignified tissues. This is a similar strategy evolved in aquatic sessile algae which are highly compliant to water flow where the drag of water is significantly higher than air. Similarly, palm trees which do not contain a vascular cambium, maintain a highly flexible, fiberous stem which also bend significantly under wind load in an avoidance mechanism as opposed to a tolerance mechanism which occurs in dicots and conifers. It is better to bend under that level of loading that try to resist. In Arabidopsis, the ephemeral nature of the floral spike in an annual plant may have evolved a mechanism to minimize the investment in structural tissues to maintain verticality and allow the stalk to flex and return to the normal upright position when the wind stops.

Line 247, delete ‘the’ in; ‘However, the none of…’

---

## [Reviewer Report]

*Comments to Author*: This article deals with the comparison between wind-induced thigmomorphogenetic reaction in Arabidopsis and that induced by a brushing treatment. Authors try to assess to what extent the brushing treatment (either unidirectional or bidirectional, with two different brushing materials) can mimic the effect of wind. A pointed in comment [6] below, the effect of brushing directionality (uni- or bi-directional) can’t be assessed because the morphology of plants differed between controls of each experiment. This is a serious problem since the title of the MS starts with “unidirectional vs. bidirectional brushing”, although experimental problems prevent the analysis of this factor. Then, this title is misleading. The other factor tested (brushing material) has no significant effect. Differences between the wind-induced and the brushing-induced reactions are noted (the change in mechanical properties is not in the same direction). Also, a positive thigmotropic reaction is evidenced.

[1] L9 and in other instances (e.g. L292): “rigid” is here opposed to “flexible”, conveying the idea that less rigid structure is more flexible and thus can bend more to reduce wind loads (by streamlining). This implicitly points to a trade-off between two strategies to cope with wind loads: being rigid to withstand wind forces (stress-resistance strategy), or being less rigid (more flexible) to avoid them (stress-avoidance strategy). However, this idea relies on some confusion between resistance and rigidity on one side, and between compliance and flexibility on the other side, and is not generally true. This confusion is common but often leads to flawed conclusions.

The mechanical properties needed for the stress-resistant strategy is ‘resistance’, i.e. being able to withstand large force without breaking. This is not the same as rigidity (measured through the elastic modulus). The rigidity (or stiffness) of a structure (or that of a material) quantifies the amount of force necessary to deform it. A material can be rigid without being resistant (e.g. glass) and can be resistant without being rigid (e.g. rubber).

The mechanical properties needed for the stress-avoidant strategy is ‘deformability’ (or flexibility). Flexibility (or deformability) denotes the properties of a material that can be bent (or deformed) a lot without breaking. Flexibility enables to reduce a lot wind loads by streamlining. This is not the inverse of rigidity. The inverse of rigidity is ‘compliance’: a more compliant material or structure need less force to be deformed, but it doesn’t mean that it can withstand a lot of deformation without breaking. A material can be flexible without being compliant (e.g. metals), and can be compliant without being flexible.

Actually, the elastic modulus E (or the bending stiffness EI) is no directly involved in the problem of wind-resistance. The modulus of rupture (MOR) is involved, as well as the deformability (that can be quantified by the ratio of MOR to E).

[2] L12-13: I don’t know the difference between “bending” and “flexing”.

[3] M&M: the presentation of the experimental design is not completely clear. As far as I understood, there are 2 sets of experiment (unidirectional / bidirectional), with 3 groups (brushing material 1, brushing material 2, control) of 20 plants each, and therefore a total of 120 plants studied. Did I understand right? The fact there are two sets of experiments should be clearly stated in the “plant material” section (currently, when you refer to “both conducted experiments”, L66-67, the is a confusion with experimental groups 1 and 2).

[4] L119: EI is not the ability of a “material” to resist bending: it is the properties of a structure, not a material property.

[5] Statistics: given the experimental design, one would expect an anova (parametric or not) to be done here, to cross-analyze the effects the brushing material and the brushing directionality.

[6] L145-150: the differences observed between the control groups of the two experiments appear to be probably highly significant. This is a serious problem. You state that the growth procedure was the same between both experiments, but the actual growth was actually very different. This problem should be pointed out in the manuscript. I don’t know what happened, but this shows that the two experiments differed not only by the brushing treatment. Because of that, you cannot (or should not) compare the effects of different brushing treatments (unidirectional vs. bidirectional).

[7] L178: “extensive” does not implies “proportional” to object size. For example, I is a extensive variable, and it depends on the 4th power of the size.

[8] L233: “an insignificant decrease” is not really a decrease… “insignificant” (not statistically significant) means that the apparent decrease is likely to be due to sampling effects. More generally, please avoid over-interpretation of non-significant results.

---

## [Reviewer Report]

*Comments to Author*: The reviewers made a number of points regarding the description of your experiments and the interpretation of your results. They are also asking for a more detailed discussion on 'stress-resistance' vs 'stress-avoidance' based on your findings. These points should be addressed in your revision.

---

## [Reviewer Report]

*Comments to Author*: This article deals with the comparison of different kinds of mechanical treatments (unidirectional brushing, bidirectional brushing, unidirectional wind) to induce thigmo-morphogenetic response in Arabidopsis. It yields two main conclusions: (1) brushing mimics the effect of wind only partly: the morphological changes induced are similar, but not the changes in mechanical properties; (2) unidirectional treatments induce a positive thigmotropic response. Both these conclusions are original findings and interesting observations that should simulate further works. The experiments have been redone since the first submission (where a significant difference occurred between controls of different treatments), and the conclusions are now strongly supported by the results. The discussion is sound in interesting. I only made two minor comments below.

[1] L116: can you specify the value of landa you used in your case?

[2] L297+: I am not so enthusiastic with the interpretation in terms of biomechanical ecological strategy (stress-avoidant strategy), for two reasons:

(i) the reasoning that concludes to a stress-avoidant strategy is based on a confusion between “flexibility” and “compliance”. A lower modulus of elasticity makes the material more compliant (i.e. less stiff), but not necessary more ‘flexible’. Flexible means that the material can be deformed at lot while not being broken. Compliant means that is deforms a lot for a given force. This is not the same. If you compare glass to steel, glass has a lower modulus of elasticity, but is also less flexible than steel.

(ii) I am not convinced that Arabidopsis inflorescence stems are really facing the danger of being broken by wind. Do you think this is a significant ecological constraint that drive selective pressure? I think the ecologically relevant mechanical perturbations of an herb is more trampling by animals and/or wind-dispersal of seeds.

---

## [Reviewer Report]

*Comments to Author*: Your revised manuscript has been reviewed by one of the initial reviewers. This reviewer made a few comments, but accepted to endorse the publication of your article.

---

## [Reviewer Report]

*Comments to Author*: Your final author version satisfactorily addressed all comments raised by the reviewer. We are pleased to accept it. Also, thank you for the LaTeX source file.